# Compressed SAR Interferometry in the Big Data Era

**Dinh Ho Tong Minh [1],* and Yen-Nhi Ngo [2]**

1   UMR TETIS, INRAE, University of Montpellier, 34090 Montpellier, France
2   Independent Researcher, 34090 Montpellier, France; ngoyennhi.ho@gmail.com
*   Correspondence: dinh.ho-tong-minh@inrae.fr

**Abstract:** Modern Synthetic Aperture Radar (SAR) missions provide an unprecedented massive interferometric SAR (InSAR) time series. The processing of the Big InSAR Data is challenging for long-term monitoring. Indeed, as most deformation phenomena develop slowly, a strategy of a processing scheme can be worked on reduced volume data sets. This paper introduces a novel ComSAR algorithm based on a compression technique for reducing computational efforts while maintaining the performance robustly. The algorithm divides the massive data into many mini-stacks and then compresses them. The compressed estimator is close to the theoretical Cramer–Rao lower bound under a realistic C-band Sentinel-1 decorrelation scenario. Both persistent and distributed scatterers (PSDS) are exploited in the ComSAR algorithm. The ComSAR performance is validated via simulation and application to Sentinel-1 data to map land subsidence of the salt mine Vauvert area, France. The proposed ComSAR yields consistently better performance when compared with the state-of-the-art PSDS technique. We make our PSDS and ComSAR algorithms as an open-source TomoSAR package. To make it more practical, we exploit other open-source projects so that people can apply our PSDS and ComSAR methods for an end-to-end processing chain. To our knowledge, TomoSAR is the first public domain tool available to jointly handle PS and DS targets.

**Keywords:** InSAR; PSI; PSDS; ComSAR; Vauvert; subsidence; TomoSAR

## 1. Introduction

Modern Synthetic Aperture Radar (SAR) satellite missions produce massive data with unprecedented characteristics [1–3]. They feature large coverages, short repeat–pass times, and high spatial resolution, opening a Big Data era for SAR data. Unlike optical satellites which capture the best images on sunny days, SAR technology takes these snapshots actively thanks to using radars, working at night, and penetrating clouds [4]. Comparing SAR images from the same position at different times can quantify surface motions with millimeters accuracy [5]. The technique is known as interferometric SAR (InSAR) [6]. Since the 2000s, InSAR time series techniques have been powerful techniques for estimating deformation from space. The principle of the InSAR time series is to take advantage of the redundancy information to minimize signal decorrelations to extract deformed signals robustly ([5,7]). While many time series InSAR methods have been developed in the last 20 years, most of them share similar characteristics so that they can be categorized into two types of techniques based on how they account for signal decorrelations. The first category of techniques is based on Distributed Scatterers (DS) ([8–10]). Distributed targets occur in natural environments (meadows, fields, bare soil) where many similarly bright scatterers contribute to the information in a resolution cell [4]. To account for signal decorrelations, it is often to select several subsets with short spatial and temporal baseline (i.e., Small Baseline Subset—SBAS) possible for analysis [8,11,12]. However, deformation measurements on distributed targets are often of lower quality and require spatial multi-looked filtering. The second approach is Permanent/Persistent Scatterers (PS) InSAR which utilizes individual scatterers dominating the signal from a resolution cell to track deformation through time. The PS interferometry (PSI) techniques provide high quality deformation information at

point target locations ([5,13–15]). However, the density of PS targets is often low, resulting in a sparse distribution of usable points. To overcome it, the advanced PSDS techniques allow us to combine both PS and DS to avoid the sparsity of identified points ([5,7,16]). The term PSDS refers nowadays to group techniques that exploit the time series phase change of both PS and DS targets.

The European Space Agency (ESA) Sentinel-1 mission allows us to have systematic SAR data based on Terrain Observation by Progressive Scans technique [17]. This leads to a massive InSAR dataset with time. This is also the same context for the next Tandem-L [18] and NISAR missions [2]. The challenge is how to exploit such Big InSAR Data for long-term monitoring [5]. Recently, there have been two initiative processing schemes to tackle DS targets (e.g., P-SBAS ([19]) and Sequential Estimator [20]). Particularly, in [20], this can be extended to work on PSDS analysis. The main objective of this work is to tackle this task by (i) developing a Compressed PSDS InSAR (so-called ComSAR) algorithm to handle the massive volume data and (ii) demonstrating it to map land subsidence of the salt mine of Vauvert area, France. The ComSAR concept had been presented at the 2021 IEEE International Geoscience and Remote Sensing Symposium IGARSS [21].

We show that the use of ComSAR to deal with SAR Sentinel-1 data offers two main advantages: (1) ComSAR can be adopted to work on massive time-series data for long-term high-precision monitoring; and (2) it yields consistently better performance when compared with the state-of-the-art PSDS InSAR and PSI techniques. Finally, we implement the PSDS and proposed ComSAR algorithms as an open-source TomoSAR package (https://github.com/DinhHoTongMinh/TomoSAR (accessed on 4 January 2022)). To our knowledge, TomoSAR is the first public domain tool available to jointly handle PS and DS targets.

## 2. PSDS InSAR: Combination of PS and DS Targets

Let us suppose that $N$ Single Look Complex (SLC) SAR acquisitions are available for a certain area of interest. The images are co-registered on a reference grid. Each pair of reference and secondary acquisitions allows the formation of an interferogram. The interferogram is the phase difference between the two complex-valued images and phase contributions due to terrain topography and orbit can be compensated. For each choice of reference acquisition, $N(N-1)/2$ interferograms can be generated from $N$ images. In practice, the estimation of parameters of interest is based on a subset or even all of these interferograms.

To form images, there are single-look and multi-look interferometric phases. For a single-look interferogram, this is the case of PS which exploits single time series pixels to calculate. For the multi-look case, the phase is the result of spatial averaging within the DS regions. The main reason for the spatially averaging phase for DS targets is that their signal-to-noise ratios (SNR) are typically low due to geometrical and temporal decorrelations [20,22,23]. By a multi-look operator, the SNR of DS can be enhanced and it can emulate PS. However, different from PS, DS phases can break the consistency of interferogram triplets (i.e., the phases of a loop of the interferograms of three images $I_1$-$I_2$, $I_2$-$I_3$, $I_3$-$I_1$ that would be zero for any point scatterer) [24,25]. This can be explained by signal decorrelation and variant systematic signals among the multi-looked interferograms. To reconstruct the consistency of the closure phase, it should employ an optimum phase estimator and a sufficient number of long time series interferograms.

For phase estimation, the goal is to extract a common-reference interferometric optimum phase series $N-1$ from all possible $N(N-1)/2$ interferograms in an SAR time series. In InSAR literature, a Maximum Likelihood Estimation (MLE) was proposed for this problem by exploiting the redundancy of the phase difference between a given pair. The retrieval of $N-1$ linked phase series from all possible interferograms is referred to as Phase Linking ([26,27]). Let us suppose that there is a similarly statistical homogeneous pixel (SHP) family available for a DS target. Its sample complex covariance matrix ($\hat{\Gamma}$) can be generated and is exploited by the phase linking technique to estimate for $N-1$ linked phase ($\hat{\lambda}$) as [20,26]:

$$\hat{\lambda} = argmax_\lambda \left\{ \Lambda^H \left( |\hat{\Gamma}|^{-1} \circ \hat{\Gamma} \right) \Lambda \right\} \tag{1}$$

where $\Lambda = exp(j\boldsymbol{\lambda})$; $^H$ is Hermitian conjugation; and $\circ$ indicates the Hadamard entry-wise product. After this procedure, the DS phase values are filtered, putting the linked phase $N-1$ phase values $\hat{\lambda}$ equal to the quality of the PS. Hence, the enhanced DS targets can be combined with PS using the same PSI algorithm.

Recently, the PSDS has been considered as a replacement for the PSI technique thanks to its ability to provide a better performance, particularly in non-urban areas [7]. Consequently, this technique is well exploited in many applications ([28–33]).

## 3. ComSAR: Compressed PSDS InSAR Algorithm

In this section, we aim to provide an approach that can reduce the data volume (and hence computational effort) and can maintain the performance.

To reduce the data volume, it is natural to use only a subset of the dataset. This is the principle of SBAS approaches. However, using the SBAS technique can lead to the risk of a phase drift from stochastic and from systematic effects ([20,34]). Consequently, this can yield large errors where the standard deviation increases with time.

To compress data volume, in the literature, principal component analysis (PCA) is a well-known technique to apply [35,36]. However, PCA fails to correctly incorporate the statistical properties of the complex covariance matrix data. This is because PCA is a geometrical rather than a probabilistic approach [36]. On the other hand, the MLE phase linking is a purely probabilistic approach, and it is well-known for estimation precise phase estimation [20,26,37]. In this way, the simple idea is to use the most coherent interferograms from their linked phases. This is mainly because the linked phases are optimum from all possible interferometric phases. Let us assume that the $N$ SAR dataset (ordered temporally) can be divided into small batches or mini-stacks with $M$ images. The compressed version $\hat{S}$ of $M$ SAR images for the $k$th sequence (see Figure 1) can be determined by a coherent summation as follows [20]:

$$\hat{S}(r,x) = \sum_{m=1}^{M} S_m(r,x)\, \xi_m \tag{2}$$

where $\xi = \frac{\hat{\Lambda}}{\|\hat{\Lambda}\|} = [\xi_1, \xi_2, \dots, \xi_M]$; $\hat{\Lambda} = exp(j\hat{\boldsymbol{\lambda}}^k)$ is the linked phase from $M$ SAR images; and $S(r,x)$ is the scene complex-valued SLC at the slant range–azimuth position.

For each mini-stack, its compression is formed using Equation (2), resulting in a strong data reduction. For example, from a stack of 300 images, we can set $M$ as 10. The processing will then estimate the 10 optimum phases by using the phase linking technique in Equation (1). These phases allow us to coherently focus the stack subset and produce a single compressed image that can represent the 10 first images of the stack. The same procedure will be repeated on the next 10 images and so on until the end of the stack, producing 30 compressed images. We note that this process has to be performed on an SHP family basis since the phase linking estimation can be valid only locally.

It is worth pointing out that, when needed, such compressed images can be used as a datum to link the history mini-stacks with the recent acquisitions and thus be able to reconstruct the full phase time series [20]. In detail, a phase linking will be performed on the compressed components ($\hat{S}$), producing a vector $\hat{\boldsymbol{\lambda}}_{cal} = \left[ \hat{\phi}_{cal}(1), \hat{\phi}_{cal}(2), \dots, \hat{\phi}_{cal}(K) \right]$ that contains the calibration phases for connecting the mini-stacks. The datum connection for the $k$th sequence is then carried out by:

$$\hat{\boldsymbol{\lambda}}_{Unified}^k = \hat{\boldsymbol{\lambda}}^k + \hat{\phi}_{cal}(k) \tag{3}$$

where the superscripts indicate the time series sequence and $\hat{\phi}_{cal}(k)$ is the $k$th of the calibration vector.

In summary, the proposed ComSAR algorithm is shown in Figure 1. Since the estimation of the linked phase is ambiguous, we set the phase of the first image in each mini-stack

to zero. Finally, PS values at these multi-reference images will be extracted from original SLCs and integrated into a compressed phase time series for the PSDS analysis.

```
% initialization
Define M number of images for a mini-stack
K = floor(N/M)

% main loop
for k = 1 to K do
    Form a mini-stack for M images
    Perform phase linking to estimate M optimum phase λ̂ᵏ
    Calculate compressed image Ŝ
end for

% combination PS and DS targets
Perform phase linking on K compressed images
Initialize PS/DS selection based on SHP properties
Assign original single-look PS values to the K images
Assign linked phase DS values to the K images

% compressed PSDS InSAR processing on K images
Refine PS/DS selection based on a phase stability criterion
Process the selected PS/DS jointly using the traditional PSI algorithm
```

**Figure 1.** ComSAR: Compressed PSDS InSAR algorithm. The algorithm divides the massive data into many mini-stacks and then compresses them.

## 4. Simulation Performances

In this section, we aim to test the proposed method for the DS targets using simulated data.

We follow a well-documented coherence model which allows us to study the behavior of temporal coherence versus time ([25,34]). In detail, the complex-valued coherence can be modeled effectively as two exponential decays and a long-term coherent component. The model can be written as follows:

$$\gamma(\Delta t) = \gamma_1 exp(j\omega_1 \Delta t)exp(-|\Delta t|/\tau_1) \\ + \gamma_2 exp(j\omega_2 \Delta t)exp(-|\Delta t|/\tau_2) + \gamma_\infty \tag{4}$$

where the term $\gamma_{1,2}$ is the initial coherence, representing the fraction of the scatterers that did not suffer from a quick decorrelation mechanism; $\omega_{1,2}$ is the characteristic phase term which is introduced to explain phase biases present in short-term interferograms; $\tau_{1,2}$ is the time constant which provides an idea of the lifetime of scatterers.

The model parameters for the C-band Sentinel-1 dataset are reported in [25]. For example, $\gamma_1 = 0.18, \gamma_2 = 0.25$; $\omega_1 = 0.03$ rad/day, $\omega_2 = 0.002$ rad/day; $\tau_1 = 11$ days, $\tau_2 = 50$ days; and $\gamma_\infty = 0.13$. Based on this, a coherence matrix is simulated for a three year time series (e.g., 6 days interval and 180 temporally ordered measurements). Each measurement contains an ensemble of 300 statistically homogeneous samples. The simulation is carried out with 1000 realizations. The root mean square error (RMSE)

(e.g., between the estimated and the simulated phases) is considered for the performance comparison.

To compare different scenarios, we consider the phase linking performance for three estimators. A Small temporal Baseline Subset (StBAS) with bandwidth 10 is equivalent to using up to lag-10 measurements at each processing level. Full bandwidth is the case that exploits all-time series data. The mini-stack is set as 10 for the proposed compression. Finally, the Cramer–Rao Lower Bound (CRLB) is the theoretical performance that uses simulated coherence for the calculation as in [26].

The performance is shown in Figure 2. The StBAS yields a large error where a phase drift is visible with time. This is because short-term coherences can often be biased. The compressed estimator outperforms other approaches by having the closest performance to the CRLB. The gain in RMSE of the compressed estimator is due to the noise-free in short-lived components.

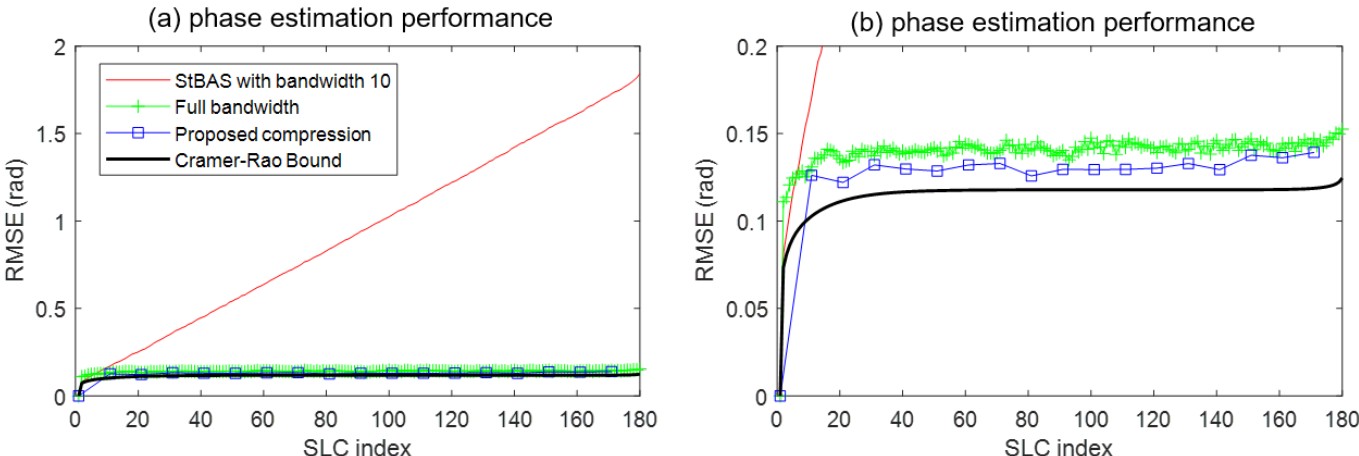

**Figure 2.** Comparison scenarios on phase linking performances for DS targets. The RMSE of phase estimation is used as a performance indicator; (**b**) is the zoom version at a different scale of (**a**) to appreciate the visualization.

## 5. Experiments with Real Data

In this section, we aim to test the proposed ComSAR method for the InSAR analysis using Sentinel-1 data for mapping subsidence in the real world in the Vauvert area, France.

### 5.1. Study Site

Vauvert area is located in Southern France (see Figure 3). The whole area is approximately 5 km × 5 km and its center is located at 4.29E longitude, 43.68N latitude. This is a salt mining industrial exploitation. The area has been injecting fresh water at high pressure into wells to dissolve salt mineral layers located about 1800–3000 m depth [38]. The extracted brine is then pumped from another several meters away. At the Vauvert area, these brine extractions have been used in the chemical industry for more than 30 years, resulting in ground motion. Hence, subsidence monitoring is necessary to avoid potential surface consequences.

In [38], the site of the salt mine of Vauvert was investigated using a stack of interferograms showing the total deformation observed between 1993 and 1999. Subsidence of about 20 mm/year and 3 km radius was reported. Recently, in [39], similar results have been observed, and the InSAR analysis has been used to increase the coverage of the leveling network.

In this work, we use the ground data from [39] as the reference which is available on request. We extracted 74 reference points over the same region for comparison with our InSAR approaches. The reference velocity varies from −20 to −2 mm/year.

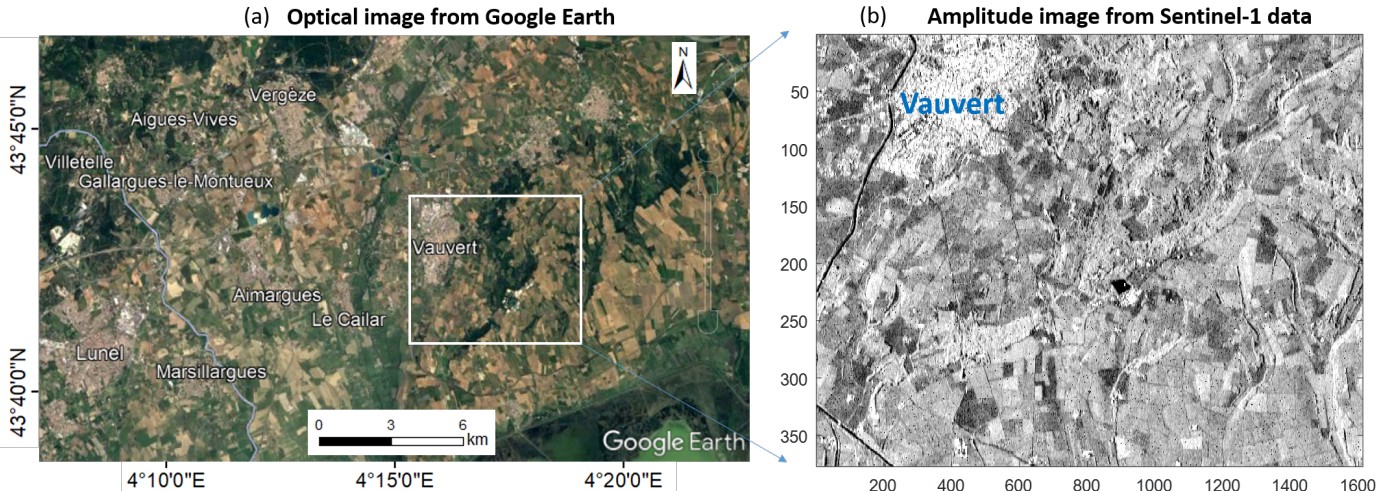

**Figure 3.** View of the Vauvert located in Southern France. (**a**) Optical Google Earth images; (**b**) amplitude image (377 pixel in azimuth and 1612 pixel in range) from the Sentinel-1 of the test site. The whole area is approximately 5 km × 5 km, and its center is located at 4.29E longitude, 43.68N latitude.

### 5.2. Processing SAR Data

For demonstration, to provide recent displacement, we processed 89 images (January 2018–December 2020) of Sentinel-1 SAR satellites (ascending track 59) using no compression (i.e., PSI and PSDS InSAR) and ComSAR techniques. They are imaged in interferometric mode (i.e., TOPS—Terrain Observation by Progressive Scans [17]). The coregistration is accomplished using two bursts of the second swath. It is worth noting that the coregistration was carefully handled to avoid a phase jump in the azimuthal direction [40]. In detail, in the azimuth direction, the accuracy of a few per-milles of a pixel is required. To meet this accuracy, the process is carried out by exploiting spectral diversity information, which considers the interferometric phase of the burst overlap regions [41]. We applied the processing over a 377 × 1612 pixel portion of the second swath of the images.

We implemented the PSDS InSAR and proposed ComSAR algorithms as an open-source TomoSAR package (https://github.com/DinhHoTongMinh/TomoSAR (accessed on 4 January 2022)) (see Appendix A). To make it more practical, we exploited the SNAP as an InSAR ([42]) processor and StaMPS ([15]) as an InSAR time series tool. For more automated processing, a SNAP-StaMPS tool can be used [43]. These tools are open source so that people can apply our PSDS and ComSAR methods for an end-to-end processing chain.

We set the small-batch as 5, resulting in 17 isolated mini-stacks. The phase linking estimation is performed on full spatial resolution in each mini-stack (see Figure 4a). The pointwise complex covariance matrices are estimated based on an ensemble of pixels in the homogenous region surrounding each pixel. For a two-sample test in DS selections, we used the Baumgartner–weiB–Schindler algorithm ([44,45]). In this way, a brotherhood of DS can be created, resulting in a family of SHPs (see Figure 4b). The DS candidate can be identified as having a number of SHP greater than a certain threshold. We set a 9 × 35 (azimuth-range direction) window to identify the number of SHP. To improve the spatial stationarity in the homogeneous region, the topographic induced phase is simulated using the 30-m resolution Shuttle Radar Topography Mission digital elevation model and compensated (i.e., subtracting the topographic phase from the original interferometric ones) prior to the coherence estimation. At each mini-stack, the data are compressed into one component. This leads to 17 images rather than 89, resulting in compression by 80 percent.

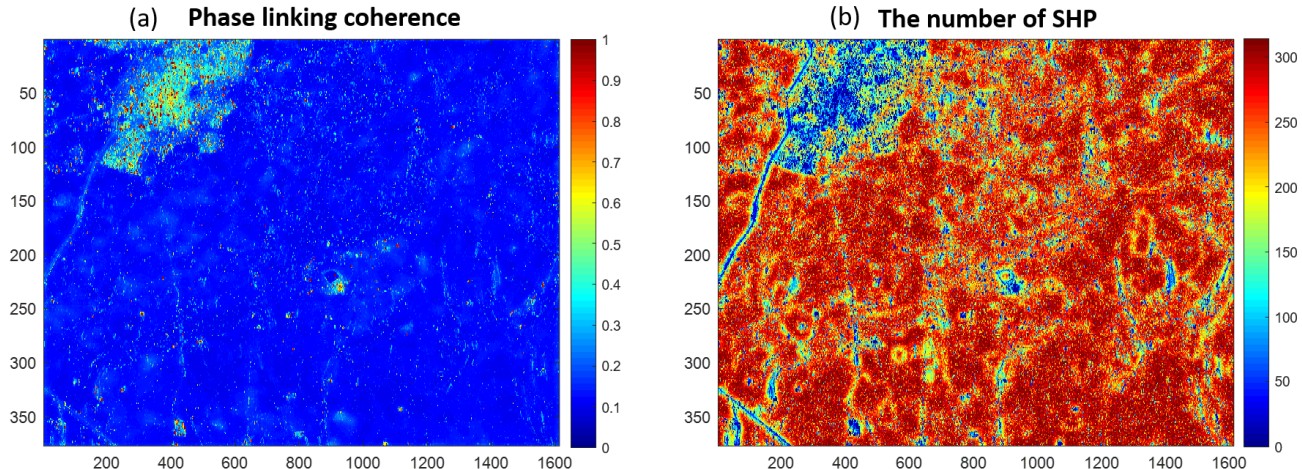

**Figure 4.** Internal results from PSDS-based processing. (**a**) the phase linking coherence corresponding to the SHP map; (**b**) the number of SHP identified using a 9 × 35 window.

### 5.3. Performance Evaluation

We consider three estimators: PSI, PSDS, and ComSAR for comparison. The PSDS and ComSAR are carried out (see Sections 2 and 3) after SNAP processing. The DS candidates are initially identified as a 20 SHP threshold and the phase linking coherence 0.25. After PSDS and ComSAR processing, the time series InSAR analysis is the same for three estimators by using StaMPS. We then identify the candidates with a threshold of 0.4 for the amplitude dispersion index criterion for three datasets. Finally, for each pixel, the phase noise standard deviation for all pixel pairs including the pixel is calculated. To drop pixels low stability, we set the minimum standard deviation as a 0.8 threshold.

Figure 5 shows the average velocity maps at PS pixels solely (Figure 5a) and PSDS points using either the PSDS InSAR (Figure 5b), or the ComSAR (Figure 5c) techniques. The distribution of the estimated velocities is shown in Figure 6. These results are relative to the average of reference data (see Figure 7a). We can observe a significant performance gain using PSDS-based approaches over the classical PSI method. Compared to the PSI algorithm based solely on PS, the increased density of PSDS pixels provides increased confidence in estimations from the advanced PSDS and ComSAR algorithms (see Table 1). It is evident that the ComSAR is of superior quality to the PSI and PSDS techniques. This is mainly because the noise component of the data space is suppressed in the data compression. In other words, in artificially compressed interferograms from the mini-stacks, the noisy short-lived components are canceled out. Therefore, these interferograms are in general higher SNR with respect to initial ones.

The reference velocity and estimated velocity are shown in Figure 7 with longitude in horizontal and latitude in a vertical direction to appreciate the ComSAR result. A circle-shaped subsidence zone is visible in Figure 7b with a 5 km diameter and subsidence up to 20 mm/year, consistent with the previous reports ([38,39]). Interestingly, to our knowledge, this is the first time we can see a very density spatial coverage in the Vauvert area from the InSAR analysis. This allows us to identify some deformation features that are difficult to detect with PS pixels.

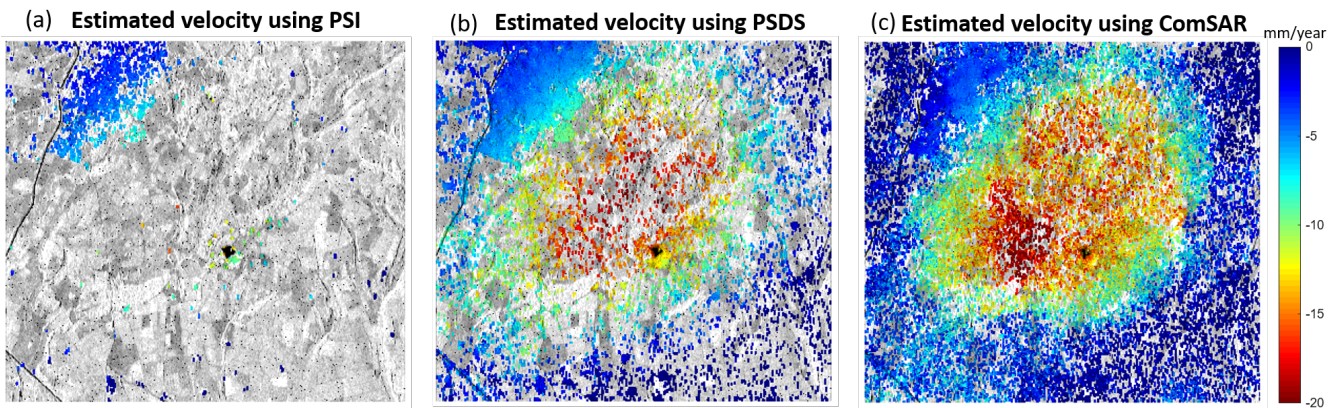

**Figure 5.** Estimated velocities at the Vauvert area, France.

**Figure 6.** Velocity histogram. The total pixels are 5100, 42,517, and 58,216 points for PSI, PSDS, and ComSAR, respectively.

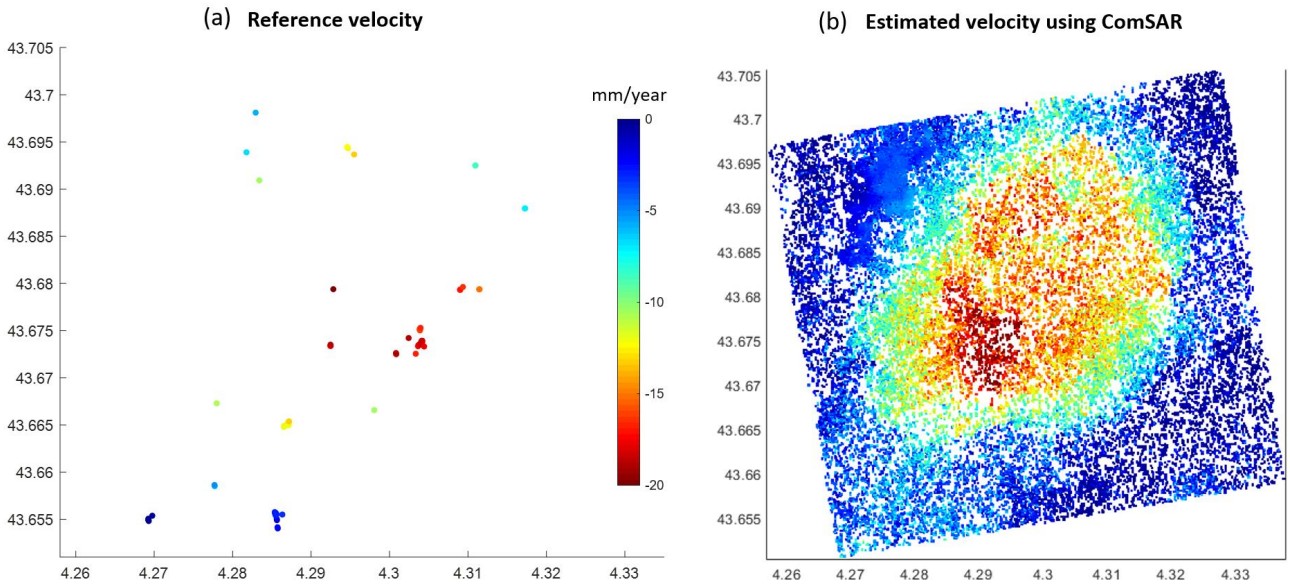

**Figure 7.** Velocities with coordination associated. (**a**) reference velocity in mm/year; (**b**) estimated velocity using ComSAR. A circle-shaped subsidence zone is visible with a 5 km diameter and subsidence up to 20 mm/year.

The comparison is shown in Table 1 and Figure 8 to quantitatively appreciate on the performance. The total size of PSDS pixels is more than 42,500 with the corresponding densities of 1700 point/km$^2$ for PSDS and 58,200 (i.e., 2328 points/km$^2$) for ComSAR. They are both higher than the PSI (i.e., 5100 points and 204 points/km$^2$) as expected. The different number of pixels in the three applied methods is due to the adding value of the DSs and the higher SNR of the compressed interferograms with respect to original ones. A very strong correlation coefficient (i.e., $R^2 > 0.8$) is found for all estimated velocities, whereas the ComSAR is the highest. The same performance is observed with the root mean square error. In terms of computation time for StaMPS processing, the PSI is the fastest as expected. For PSDS, it takes 168 min to process, whereas only 25 min is required for ComSAR. Thus, instead of processing 89 images, we work on 17 images and get even better performance.

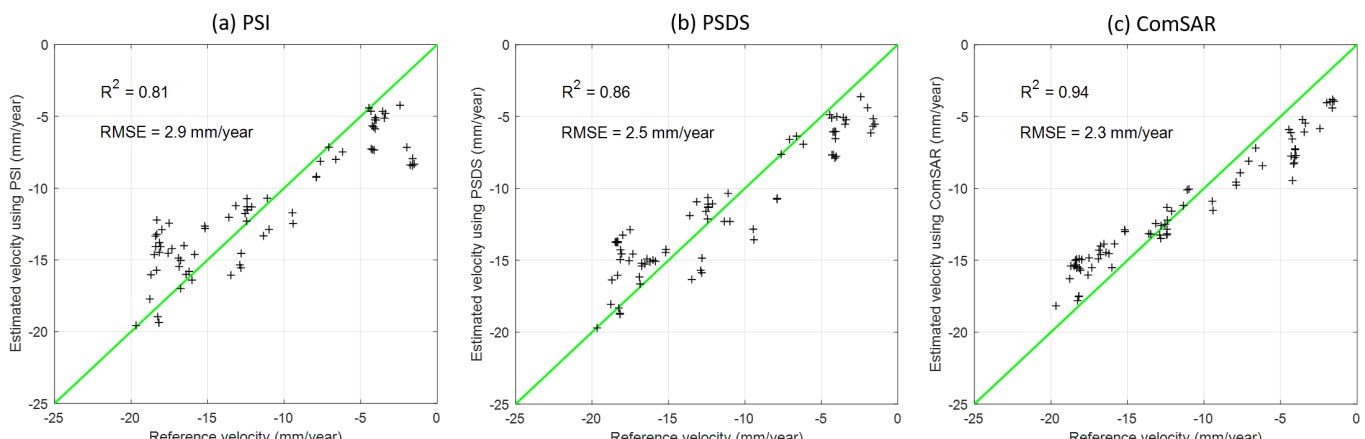

**Figure 8.** The cross-plot 1:1 velocity comparison. (**a**) estimated velocity using PSI; (**b**) estimated velocity using PSDS; (**c**) estimated velocity using ComSAR.

**Table 1.** Characteristics of performance.

| Parametes | PSI | PSDS | ComSAR |
|---|---|---|---|
| Total image | 89 | 89 | 17 |
| Total point | 5100 | 42,517 | 58,216 |
| Density (point/km$^2$) | 204 | 1700 | 2328 |
| Duration (minute) | 8 | 168 | 25 |
| Coefficient R$^2$ | 0.81 | 0.86 | 0.94 |
| RMSE (mm/year) | 2.9 | 2.5 | 2.3 |

## 6. Discussion

This work introduces a novel ComSAR algorithm based on a compression technique for reducing computational efforts. We obtained good results and maintained the performance robustly while working on compressed versions. First, the ComSAR performance is validated via simulation, showing that the compressed estimator is close to the theoretical Cramer–Rao lower bound under a realistic C-band Sentinel-1 decorrelation scenario. Both persistent and distributed scatterers are exploited in the ComSAR algorithm. We demonstrated its application to Sentinel-1 data to map land subsidence of the salt mine Vauvert area, France. We provided a very density spatial coverage in the Vauvert area, identifying some deformation features that are difficult to detect with PS pixels. The proposed ComSAR yields consistently better performance when compared with the state-of-the-art PSDS and PSI techniques. Finally, to make it more practical, we make our PSDS and ComSAR algorithms as an open-source TomoSAR package. To our knowledge, TomoSAR is the first public domain tool available to jointly handle PS and DS targets.

We show that the compressed estimator outperforms other approaches in both simulation and experiments with Sentinel-1 data. The gains of the compressed estimator are due to the fact that the coherent integration within the mini-stacks attenuates both noise and short-lived components [46]. In other words, the noise component of the data space is canceled out in the data compression, resulting in a better SNR of the compressed interferograms with respect to the original ones.

Concerning the short temporal baselines, the result (see Section 4) yields a large error with time due to short-term coherences, which can be often biased. The problem is related to what is discussed in [25] about a systematic bias. Although it is noted that short temporal baselines can gain the reduction in temporal decorrelation, the coherence should be considered at a rather long (i.e., more than one month as suggested in [25]) time distance to decrease the overall phase error.

It is worth noting that, since ComSAR requires a mini-stack for compression, it works best as most deformation phenomena develop slowly. For areas that are undergoing complicated deformation (e.g., earthquake and volcano), ComSAR can be used as a datum to reconstruct the full phase time series (see Equation (3)). Furthermore, for the proposed procedure to work, it is essential that some coherent components must exist in mini-stacks. If it is not the case, the interferometric phases between the compressed images can be noisy and no information will be retrieved due to decorrelation. This is particularly true in complex motion patterns. To better understand it, future researchers should consider two-dimensional simulations to better take into account these motion patterns.

## 7. Conclusions

This paper has introduced the ComSAR (which considers both PS and DS targets) processing scheme to exploit the unprecedented Big InSAR Data. The compressed estimator for DS targets can yield better performance for the full bandwidth covariance algorithm for a realistic Sentinel-1 decorrelation scenario. The compressed interferograms are higher SNR with respect to original ones because noisy short-lived components are canceled out. We demonstrated the ComSAR technique on mapping subsidence at the salt mine Vauvert area. ComSAR yields consistently better performance when compared with the PSI and

PSDS techniques, providing a promising tool for applications in long-term high-precision monitoring.

We implemented the PSDS and proposed ComSAR algorithms as the open-source TomoSAR package. To make it more practical, we exploited the SNAP as an InSAR processor and StaMPS as a time series tool. These projects are open source so that people can apply our PSDS and ComSAR methods for an end-to-end processing chain. To our knowledge, TomoSAR is the first public domain tool available to jointly process PS and DS targets.

The ComSAR algorithm is suited to most deformation phenomena that develop slowly. For areas that are undergoing complex motion patterns, more research is needed to characterize the limitation of the ComSAR. For instant, two-dimensional simulations can be used to account for these motion patterns in future studies.

The evolution of new SAR satellites, such as Sentinel-1C and Sentinel-1D, allows enabling the continuity of the Big Data. In future research, more studies are needed to improve algorithms that can be possible to carry out not only in the long-term but also in large-scale applications.

**Author Contributions:** Conceptualization: D.H.T.M. and Y.-N.N.; visualization: D.H.T.M. and Y.-N.N.; writing—original draft: D.H.T.M. and Y.-N.N.; editing: D.H.T.M. and Y.-N.N. All authors read and approved the final manuscript.

**Funding:** The ComSAR work was supported in part by the Centre National d'Etudes Spatiales/Terre, Ocean, Surfaces Continentales, Atmosphere (CNES/TOSCA) (Project MekongInSAR and BIOMASS-valorisation), UMR TETIS, and Institut national de recherche en agriculture, alimentation et environnement (INRAE).

**Institutional Review Board Statement:** Not applicable.

**Informed Consent Statement:** Not applicable.

**Data Availability Statement:** Sentinel-1 data are freely provided by the European Space Agency.

**Acknowledgments:** We thank Séverine Liora Furst from Université Savoie Mont-Blanc for sharing the Vauvert reference data.

**Conflicts of Interest:** The authors declare no conflict of interest.

## Appendix A. Appendix on TomoSAR

The TomoSAR was designed as an end-to-end software platform for processing multi-sensor SAR images [47]. The kernel of this platform supports the entire processing from SAR, Interferometry, Polarimetry, to Tomography (so-called TomoSAR). The platform is based on the previous works done by D. Ho Tong Minh in the frame of his research at the Politecnico di Milano ([48]), CESBIO ([49]) and continuously developed by his group at INRAE ([21,47,50]). We made it open-source on a part of TomoSAR for PSDS and ComSAR algorithms (https://github.com/DinhHoTongMinh/TomoSAR (accessed on 4 January 2022)). Finally, we created an interactive social forum (https://www.facebook.com/groups/RadarInterferometry (accessed on 4 January 2022)) where researchers can consult other colleagues, share information, help each other, and develop topics about not only our open-source TomoSAR but also everything in the InSAR field.

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
