# Peer review of "Compressed SAR Interferometry in the Big Data Era"

_remotesensing, doi:10.3390/rs14020390_

Round 1

Reviewer 1 Report

Authors propose some peculiar idea to reduce number of computations when working with huge data stacks (via data compression) when  solving the task of the displacements detection by means of multitemporal SAR interferometry. There are several comments that are caused probably  by an insufficiently clear and precise presentation of the material.

  1. Leading idea of authors is to select most coherent interferograms (see line 126) “In this way, the simple idea is to use the most coherent interferograms from their linked phases”. They did not explain how to choose them from a set of available ones.
  2. Line 86. Authors made incorrect statement: "By multi-look operator, the SNR of DS can be enhanced and it can become PS”. Multilooking in a given case improves SNR of DS phase. Also DS cannot become PS, but it can be treated as PS.
  3. Expression (1) authors use to describe the way to get linked phases is incorrect (see for example [14] or [18].
  4. Authors used Sentinel-1 SAR data, did not mention SAR operation mode. I think. In the TOPS mode of operation DS approach they used will not work as precise coregistration of pixels is required.

Author Response

Authors propose some peculiar idea to reduce number of computations when working with huge data stacks (via data compression) when solving the task of the displacements detection by means of multitemporal SAR interferometry. There are several comments that are caused probably by an insufficiently clear and precise presentation of the material.

  1. Leading idea of authors is to select most coherent interferograms (see line 126) “In this way, the simple idea is to use the most coherent interferograms from their linked phases”. They did not explain how to choose them from a set of available ones.

We agree to add more information to better explanation. This work shows that the gains obtainable by the algorithm phase linking. Since linked phases are the N-1 phase optimum from N(N-1)/2 possible phase, by exploiting linked phase, we can form an optimum one. We added it in the text: ‘This is mainly because the linked phases are optimum from all possible interferometric phases’.  

  1. Line 86. Authors made incorrect statement: "By multi-look operator, the SNR of DS can be enhanced and it can become PS”. Multilooking in a given case improves SNR of DS phase. Also DS cannot become PS, but it can be treated as PS.

We agree to correct this. We changed the phrase to ‘By multi-look operator, the SNR of DS can be enhanced and it can emulate PS.’

  1. Expression (1) authors use to describe the way to get linked phases is incorrect (see for example [14] or [18].

We tried to double check our expression (1) and the notion in equation 2 of [14]. In our expression,  and are used for complex matrix and its modulus. In equation 2 of [14],  and  are used for complex matrix and its modulus. It is the same way to describe the estimation of linked phase indeed. We cited [14] and [18] to support this expression.     

  1. Authors used Sentinel-1 SAR data, did not mention SAR operation mode. I think. In the TOPS mode of operation DS approach they used will not work as precise coregistration of pixels is required.

We agreed to add the information on TOPS mode for the Sentinel-1 data. For TOPS mode, the accuracy of a few per-mille of a pixel in the azimuth direction is required for phase analysis. We added it in the text as follow:  They are acquired in interferometric mode (i.e., TOPS - Terrain Observation by Progressive Scans [17]). The coregistration is accomplished using 2 bursts of the second swath. It is worth noting that the coregistration was carefully handled to avoid a phase jump in the azimuthal direction [40]. In detail, in the azimuth direction, the accuracy of a few per-mille of a pixel is required. To meet this accuracy, the process is carried out by exploiting spectral diversity information which considers the interferometric phase of the burst overlap regions [41].

Reviewer 2 Report

The article presents a methodology to reduce the number of interferograms in the interferometric processes (PS, DS, PSDS). Furthermore, it is reported that the quality of the interferograms is improved by the compression process. The work is very interesting, since with the large number of SAR images available today, and more in the future, these types of techniques are very important to reduce computational costs in large interferometric projects.

The quality of the presentation of the work is very high, and I recommend its publication after some clarification and improvements in its content:

1. In section 3 the problem of using short temporal baselines in interferograms (case of SBAS) is discussed. But it should also be noted the reduction in temporal decorrelation, and depending on the type of terrain in the area of ​​interest, it may be advisable to opt for shorter temporal baselines.

2. I think there is an error in the formulation of line 130. Please check it.

3. There is a concept that is not clarified in the work, and it is how to reconstruct the time series of the M original images from the processing of the K compressed images. Could you explain this final step in detail?

4. Finally, the possible limitation of ComSAR with certain complex motion patterns is discussed. Could you explain in detail what these limitations consist of? Especially in the case of rapid movements and/or with non-linear patterns, since in many cases the interferometric techniques have to face this type of problem.

Author Response

The article presents a methodology to reduce the number of interferograms in the interferometric processes (PS, DS, PSDS). Furthermore, it is reported that the quality of the interferograms is improved by the compression process. The work is very interesting, since with the large number of SAR images available today, and more in the future, these types of techniques are very important to reduce computational costs in large interferometric projects.

The quality of the presentation of the work is very high, and I recommend its publication after some clarification and improvements in its content:

  1. In section 3 the problem of using short temporal baselines in interferograms (case of SBAS) is discussed. But it should also be noted the reduction in temporal decorrelation, and depending on the type of terrain in the area of ​​interest, it may be advisable to opt for shorter temporal baselines.

Thank you very much for your suggestion. We added this interpretation in section Discussion.

Concerning the short temporal baselines, the result (see section 4) yields a large error with time due to short-term coherences which can be often biased. The problem is related to what is discussed in [47] about a systematic bias. Although it is noted that short temporal baselines can gain the reduction in temporal decorrelation, the coherence should be considered at a rather long (i.e., more than one month as suggested in [47]) time distance to decrease the overall phase error.

  1. I think there is an error in the formulation of line 130. Please check it.

Thank you! It should have a norm in the denominator. We fixed it.

  1. There is a concept that is not clarified in the work, and it is how to reconstruct the time series of the M original images from the processing of the K compressed images. Could you explain this final step in detail?

We agree to add more information and equation 3 to better explanation it in the text.

In detail, a phase linking will be performed on the compressed components, producing a vector that contains the calibration phases for connecting the mini-stacks. The datum connection for the kth sequence is then carried out.

  1. Finally, the possible limitation of ComSAR with certain complex motion patterns is discussed. Could you explain in detail what these limitations consist of? Especially in the case of rapid movements and/or with non-linear patterns, since in many cases the interferometric techniques have to face this type of problem.

This is an interesting question and it should be the next logical step. We think the answer can be drawn when we take into account two-dimensional simulation. We keep these works in future effort. We added it in the Discussion.

Furthermore, for the proposed procedure to work, it is essential that some coherent components must exist in mini-stacks. If it is not the case, the interferometric phases between the compressed images can be noisy and no information will be retrieved due to decorrelation. This is particularly true in complex motion patterns. To better understand it, future researchers should consider two-dimensional simulations to better take into account these motion patterns.      

Reviewer 3 Report

Dear Authors,

This is very interesting work, and sounds very promising for the new challenges that all researchers are facing with the vast amount of available SAR datasets. There are two main issues with your manuscript:

1) The length is quite short to be considered as an Article (read Instructions for Authors) and also a separate Discussion section is missing. You need to enlarge the Introduction and Conclusions sections, and create a new Discussion section.

2) There are 11 self-citations of the corresponding author (6 of them in the Appendix). Although I realise that some are crucial to support your work, the others seemed as unnecessary. You need to remove and keep only the most important.

Other than that, your figures and table need re-arranging as it is difficult to follow them in your text.

My detailed comments can be found in the attached pdf.

Author Response

This is very interesting work, and sounds very promising for the new challenges that all researchers are facing with the vast amount of available SAR datasets. There are two main issues with your manuscript:

1) The length is quite short to be considered as an Article (read Instructions for Authors) and also a separate Discussion section is missing. You need to enlarge the Introduction and Conclusions sections, and create a new Discussion section.

Thank you very much. We agree to add a new Discussion section and enlarge the Introduction and Conclusions sections. We added these improvements in the text accordingly.

 2) There are 11 self-citations of the corresponding author (6 of them in the Appendix). Although I realise that some are crucial to support your work, the others seemed as unnecessary. You need to remove and keep only the most important.

We agree to remove and keep the important ones in the Appendix.

Other than that, your figures and table need re-arranging as it is difficult to follow them in your text.

We tried to rearrange the figures and table to better follow them in the text.

My detailed comments can be found in the attached pdf.

Thank you very much for your suggestions.

This ComSAR work is mainly on the methodology. The Vauvert area is just for demonstration. We believe the current title is fit to this context.

The text is based on our conference paper which is not a peer-review article. We updated it to meet the article requirement.      

We report figure 4 and 5 in an image coordination system rather than WGS84 for convenient and better visualization.   

Suggestions on reference and rephrases were updated accordingly. Thank you very much.  

Round 2

Reviewer 3 Report

Dear authors,

Thank you very much for replying to some of my comments. The introduction section was significantly improved, and the newly added discussion section serves its purpose.

However, you did not address all of my comments/suggestions. The first 10 lines of the Abstract must be paraphrased as they are identical to those in your conference paper. The issues I brought to your attention during the previous round of reviews, i.e. the manuscript size, the outdated references and the number of self citations, although reduced from 11 to 7, still remain.

Additionally, figures 5-8 and table 1, located in pages 9 and 10, need to be incorporated within the text, as it still difficult for the reader to follow. Finally, the newly added text needs to be edited for English language.